# Genomic characterization of a severe West Nile Virus transmission season using a single reaction amplicon sequencing approach

Shawn Freed Jr.[1,2☯], Sarah Chandler[3☯], Sarah Uhm[1☯], Zach Pella[1,2], Dikchha Gurung[1], Hallie Smith[4], Tammy Dowdy[4], Amanda M. Bartling[5], Ava Butz[1], Michael R. Wiley[5,6], M. Jana Broadhurst[6], Sydney Stein[4], Emily L. McCutchen[5], Jeff Hamik[4], Peter C. Iwen[5,6], Nick Downey[7], Kaylee S. Herzog[1], Joseph R. Fauver[1]*

1 University of Nebraska Medical Center, College of Public Health, Department of Epidemiology, Omaha, Nebraska, United States of America, 2 Association of Public Health Laboratories, Bethesda, Maryland, United States of America, 3 University of Nebraska Medical Center, College of Medicine, Omaha, Nebraska, United States of America, 4 Nebraska Department of Health and Human Services, Lincoln, Nebraska, United States of America, 5 Nebraska Public Health Laboratory, Omaha, Nebraska, United States of America, 6 University of Nebraska Medical Center, College of Medicine, Department of Pathology, Microbiology, and Immunology, Omaha, Nebraska, United States of America, 7 Integrated DNA Technologies, Coralville, Iowa, United States of America

☯ Authors contributed equally to this work.
* jfauver@unmc.edu

## Abstract

West Nile virus (WNV) is an endemic arthropod-borne virus that has routinely caused seasonal outbreaks in the United States since it was first detected in 1999. While phylogenetic studies have shown how WNV has diversified and undergone genotype replacement since introduction, more geographically focused studies are needed to understand intricate transmission dynamics at local and regional scales. In this study, we validate the IDT xGen WNV panel, a novel single reaction amplicon-based Next-Generation Sequencing approach, to generate high-quality WNV genomes and compare it to the "Primal Scheme" assay for WNV, a common amplicon sequencing strategy. By generating >250 genomes from mosquito pools, we show that the IDT xGen WNV panel generated coding-complete and accurate WNV genomes when compared to the current sequencing approaches. Additionally, we used this approach to generate 100 coding-complete WNV genomes from surveillance pools of mosquitoes collected in Nebraska during the 2023 outbreak. Our discrete phylogeographic analysis revealed substantial genetic diversity in WNV genomes from 2023 with minimal clustering across the state. This study demonstrated the utility of a single reaction amplicon-based sequencing approach to generate quality WNV genomes from routine surveillance samples and characterize WNV transmission dynamics in a high-incidence setting.

**Data availability statement:** All data generated this study are available at the European Nucleotide Archive (ENA) under accession PRJEB96341. Final genomes and read-level data are available for each sample. These data have been cross listed on NCBI GenBank and NCBI Short Read Archive and can be found via specific accession numbers for each sample available on our GitHub Page. All code and metadata, including accession numbers, used in this analysis are available on our GitHub: github.com/josephfauver/ WNV_Methods_Outbreak_Manuscript.

**Funding:** This work was supported in part by start-up funds from the UNMC VCR Office provided to JRF. Additionally, this publication is supported by Cooperative Agreement Number NU60OE000104 (CFDA #93.322), funded by the Centers for Disease Control and Prevention (CDC) of the US Department of Health and Human Services (HHS) providing salary support to SF and ZP. Its contents are solely the responsibility of the authors and do not necessarily represent the official views of APHL, CDC, HHS or the US Government. The funders had no role in study design, data collection and analysis, decision to publish, or preparation of the manuscript.

**Competing interests:** I have read the journal's policy and the authors of this manuscript have the following competing interests: ND is an employee of IDT. We confirm that these competing interests will not alter adherence to PLOS policies on sharing data and materials. All other authors declared that no competing interests exist.

## Author summary

West Nile virus (WNV) remains the most impactful arthropod-borne virus in the United States. Genomic epidemiology of WNV can inform our understanding of virus emergence and transmission, ultimately informing mosquito surveillance and control programs. In this study, we validated a simple, single-reaction amplicon sequencing approach by generating >250 coding-complete WNV genomes from WNV positive pools of mosquitoes. We then characterized the 2023 transmission in Nebraska, a high-incidence state, and showed that the western portion of the state saw early and intense WNV transmission, as evidenced by high infection rates in both mosquitoes and humans. We showed there was significant genetic diversity in WNV genotypes with minimal clustering of WNV genomes across the state, indicating the early transmission initiation in the western portion did not seed transmission in other areas of the state. Our study demonstrated the utility of a single-reaction sequencing approach to generate high-quality WNV genomes directly from pools of mosquitoes collected for surveillance purposes while also highlighting the substantial WNV genetic diversity present during a severe WNV transmission season.

## Introduction

West Nile virus (WNV) is an arthropod-borne flavivirus (arbovirus) that exists in a transmission cycle between passerine birds and *Culex* species mosquitoes and is widely distributed across the United States [1]. Since introduction to the United States, there have been >60,000 diagnosed clinical cases of WNV disease, of which >30,000 have resulted in severe neuroinvasive disease [2]. Importantly, clinical cases are underreported and the true burden of infections in the United States is estimated to be in the millions [3]. A substantial portion of individuals who develop symptomatic disease experience neurological sequelae [4], underscoring WNV as a significant source of both acute and chronic disease. As there is no licensed human vaccine or specific treatment for WNV, control depends entirely on reducing exposure to infectious mosquitoes [5]. Therefore, understanding WNV transmission dynamics is imperative to determine risk of human infection and ultimately implement control measures [6]. Entomological surveillance provides the best spatial-temporal indication of human risk of WNV transmission [7–9]. Accordingly, vector mosquitoes are systematically collected and tested for the presence of WNV seasonally throughout the United States [10]. While informative for inferring contemporary risk of infection, a simple presence/absence measure of WNV in mosquito populations does not allow for broader inference of transmission patterns. Understanding the connectivity of WNV populations across spatial scales, genotype maintenance through and across transmission seasons, and patterns of emergence and transmission initiation requires the addition of WNV genomic information.

Numerous efforts to understand WNV transmission patterns and evolution at a national scale have been undertaken using genomic approaches. These studies have

shown that WNV dispersed rapidly across the United States during the early epidemic period [11,12] and dissemination was characterized by long-range movements of the virus [13]. WNV populations are thought to be loosely structured by geography across the US with substantial circulating genetic diversity in concurrent areas of transmission [11,14]. Since the introduction of WNV, multiple genotypes have arisen and come to dominate transmission. This includes the replacement of NY99 by WN02 [15,16], the rise of SW03 [17] in the Southwest, and more recently the emergence of NY10 [18] in the Northeast, Southeast, and Midwest regions of the United States. While national studies are crucial for inferring large-scale transmission patterns, such as the emergence of new genotypes, WNV transmission is highly heterogeneous across the United States. Studies on local WNV dynamics can reveal important factors influencing transmission and inform risk. Longitudinal genomic analysis of WNV in California identified the co-circulation of WN02 and SW03 and implicated overwintering mosquitoes and non-migratory bird populations for maintaining WNV genotypes through seasons [19]. An analysis of WNV genomes sampled over a decade in Connecticut indicated that control measures were unlikely to have a lasting impact beyond a disease season due to repeated WNV introductions that reinitiate transmission year to year [20]. Additionally, longitudinal genomic analysis and virus characterization from New York state identified the emergence of a novel genotype, NY10, and found its defining mutations resulted in increased infectivity and transmissibility by *Culex pipiens* mosquitoes and led to NY10 becoming the dominant genotype across the state and elsewhere in the United States [18,21,22]. Prior studies of WNV dynamics at local levels reveal important factors influencing transmission and potential risk. While informative, genomic studies have been limited to a few states and localities across the United States and have not included areas with the highest disease burden, such as the Central Plains [1].

Untargeted Next-Generation Sequencing (NGS) strategies, such as shotgun metagenomics, from pools of mosquitoes would be an ideal sequencing approach as it would not require prior knowledge of the pathogen, however these approaches lack sensitivity and result in a high proportion of non-arbovirus read data [23]. Therefore, sequencing arbovirus genomes from entomological samples requires either the isolation of virus in cell culture prior to sequencing and/or using target-specific amplification approaches. While isolating WNV in cell culture increases the specificity of NGS data, most mosquito surveillance programs around the United States rely on molecular detection methods, specifically RT-qPCR, to identify WNV in mosquito populations and no longer isolate viruses using cell culture [10]. Multiple amplicon-based sequencing approaches have been developed to generate high-quality WNV genomes [22,24] however these approaches rely on single amplicons to cover each region of the genome [24,25], which can result in amplicon drop-out when mutations occur in amplicon primer binding regions [26]. In this study, we described the development and validation of a single reaction amplicon-based NGS approach to generate coding-complete WNV genomes, called IDT xGen WNV. We compared our approach to the "Primal Scheme" amplicon-sequencing strategy that is commonly used for sequencing WNV. Finally, we demonstrated the utility of this approach by characterizing the 2023 WNV transmission season in Nebraska using entomological, epidemiological, and genomic data. We have shown that the IDT xGen WNV panel produced coding-complete and accurate genomes ideal for phylogenetic inference and outbreak characterization.

## Materials and methods

### IDT xGen WNV panel development

The IDT xGen Amplicon sequencing approach is similar to Primal Scheme, however it differs in that Primal Scheme uses an overlapping primer approach to generate amplicons to the complete virus genome by staggering primer design and conducting PCR in two separate reactions, where the IDT xGen Amplicon assay makes uses of "super amplicons" that are elongated and span multiple primer sequences, thus allowing for PCR to be conducted in a single tube (**S1 Fig**). A custom IDT xGen Amplicon panel was designed in partnership with Integrated DNA Technologies (IDT, Coralville, Iowa). A total of 50 publicly available WNV genomes were downloaded from GenBank [27] to create a multisequence alignment (MSA) using CLUSTAL Omega [28]. Primer sequences were designed from the consensus sequence to capture contemporary WNV Lineage 1A diversity while being robust to amplicon dropout due to nucleotide mismatches in primer binding regions. Primer sequences span the complete coding-sequence of WNV and do not amplify the 5' and 3' untranslated regions.

## Sample selection

All WNV genomes in this study were generated from WNV positive mosquito pools collected by the Nebraska Department of Health and Human Services (NeDHHS) for routine seasonal WNV surveillance purposes from 2012-2024. Briefly, CDC light traps are placed in multiple counties across the state to collect *Culex* species mosquitoes from May through September. Trap contents are sent to NeDHHS on dry ice for morphological species identification and pooling. Pools are separated by location, date, and species and up to 50 individual mosquitoes stored in a tube with PBS prior to WNV testing. Mosquito pools were deemed positive for WNV via RT-qPCR [29] performed by the Nebraska Public Health Laboratory (NPHL). To characterize genomic data generated with the IDT xGen WNV panel, a total of 287 WNV positive mosquito pools collected during 2012–2024 were selected for sequencing. Between 20–50 WNV positive mosquitoes pools were selected for each year in the study in an attempt to get geographic and temporal representation across the state. This sample set included 107 samples from the 2023 WNV transmission season in Nebraska. Additionally, a subset of 30 samples were sequenced using both the IDT xGen WNV panel and the WNV Primal Scheme assay for direct comparison [24].

## IDT xGen WNV library preparation, sequencing, and consensus sequence generation

Prior to nucleic acid extraction, all mosquito pools were homogenized by placing a single ball bearing in the tube containing mosquitoes/PBS and placed on a tissue homogenizer at 25 Hz for 1 minute. For WNV positive mosquito pools collected during 2012–2020, total nucleic acid was extracted using the MagMAX Viral/Pathogen Nucleic Acid Isolation Kit (ThermoFisher) using a KingFisher Flex instrument (ThermoFisher) according to manufacturer's instructions. For WNV positive pools collected during 2021–2024, RNA was provided by NPHL. RNA was retested for the presence of WNV via a specific RT-qPCR assay using the Luna Universal Probe One-Step RT-qPCR Kit (New England Biolabs [NEB]) as previously described [30]. RNA was reverse transcribed to create cDNA using LunaScript RT SuperMix (NEB) according to manufacturer's instructions and 10µl was used as input to the IDT xGen Amplicon Core Kit (IDT). Total RNA or cDNA was not quantified prior to library preparation. Library preparation was conducted according to manufacturer's instruction with the following modifications: increased elution volume following panel-specific multiplex PCR to 25µl, increased elution post indexing PCR to 20µl, and removed samples from beads in each clean-up step. Samples were indexed with IDT xGen Normalase UDI Primers. Samples were normalized and pooled with the IDT xGen Normalase module according to manufacturer's instructions. Following library normalization and pooling, final library pools were quantified with the Qubit dsDNA Quantification HS Assay (ThermoFisher). An initial pool of 20 samples was sequenced on the Illumina MiniSeq sequencing system at the University of Nebraska Medical Center (UNMC), and the remaining pools were sent to the Yale Center for Genomic Analysis (YCGA) for sequencing on Illumina NovaSeq 6000 sequencing system. A total of 100,000 reads were targeted for each sample. Following sequencing and demultiplexing, data was transferred to the Holland Computing Center (HCC) at the University of Nebraska for quality control and consensus sequence generation. Read level data were assessed for quality and had Illumina adapter sequences trimmed with fastp [31] where reads with a Q score <15 were removed. Following quality control, reads were aligned to the previously described consensus WNV genome using BWA [32], primer sequences were trimmed with fgbio [33],.bam files were created and indexed with samtools [34], and consensus sequences were generated with ViralConsensus [35]. At least 10 reads (10x depth) were required to call a base in the consensus sequence, and regions of the genome with <10 reads were assigned the ambiguous nucleotide "N". A detailed bioinformatic script to conduct QC and consensus sequence generation from the IDT xGen WNV panel can be found in our GitHub repository.

Additionally, a subset of 10 samples prepared for Illumina sequencing were also prepared for sequencing on the Oxford Nanopore Technologies (ONT) MinION sequencing platform. These samples were processed and analyzed as described above using a modified version of the primer panel and of the IDT xGen Amplicon Core Kit reagents. The bioinformatic analysis is similar to that described above, however a constrained sequence length between 200–2000 base pairs was implemented using seqkit [36], and reads were aligned to the reference genome using minimap2 [37]. A detailed

bioinformatic script to conduct QC and consensus sequence generation from the IDT xGen WNV panel sequenced on the ONT MinION platform can be found in our GitHub repository.

## Primal scheme library preparation, sequencing, and consensus sequence generation

The subset of RNA from a total of 30 WNV positive mosquito pools described above also underwent library preparation using the Primal Scheme assay designed for WNV [24,25]. Following cDNA synthesis as described above, 5μl of cDNA was aliquoted to each of two tubes for multiplex PCR with Pool 1 and Pool 2 primers, respectively. Multiplex PCR was performed as previously described [24] with reduced PCR cycles to 30. Following bead-based sample purification using KAPA Pure Beads (Roche), samples were eluted into 25ul molecular grade water. Pool 1 and Pool 2 contents were pooled by sample, quantified with the Qubit dsDNA Quantification HS Assay (ThermoFisher), and used as input for library preparation using the Illumina DNA Prep Kit (Illumina) according to manufacturer's instructions. Following library preparation, samples were quantified as previously described and pooled to equal concentrations prior to sequencing. Initially, 10 samples were sequenced on the MiniSeq system at UNMC. An additional pool of the remaining 20 samples was sent to YCGA for sequencing on Illumina NovaSeq 6000 sequencing system. A total of 100,000 reads were targeted for each sample. Following sequencing and demultiplexing, data was transferred to the HCC at the University of Nebraska for quality control and consensus sequence generation. Read level data were assessed for quality and had Illumina adapter sequences trimmed using fastp [31] where reads with a Q score <15 were removed. Following quality control, reads were aligned to a reference WNV genome using BWA [32],.bam files were created and indexed with samtools [34], and primers were trimmed and consensus sequences were generated using iVar [24]. At least 10 reads (10x depth) were required to call a base in the consensus sequence, and regions of the genome with <10 reads were assigned the ambiguous nucleotide "N". A detailed bioinformatic script to conduct QC and consensus sequence generation can be found in our GitHub repository.

## Validation of IDT xGen WNV panel

WNV genomes generated using the IDT xGen WNV panel were assessed for genome completeness by determining the number of ambiguous nucleotides ("N") in the genome assembly using a custom bash script. The percent of reads aligning to the WNV genome was determined by parsing the output of the samtools flagstat command incorporated into our bioinformatic pipeline. The temporality of WNV genomes was assessed using TempEst [38]. Genomes generated with the IDT xGen WNV panel were aligned using MAFFT [39] and a maximum-likelihood (ML) phylogenetic analysis was conducted using PhyML [40]. The ML tree and a metadata file containing sample collection dates were imported into TempEst to calculate substitutions per site, correlation coefficient, and residual means squared. To compare genome accuracy, a multisequence alignment was generated using MAFFT [39] of the 30 samples prepared and sequenced with both the IDT xGen WNV panel and the Primal Scheme WNV assay (60 sequences total). A distance matrix for the alignment was produced in Geneious [41] and nucleotide differences between consensus sequence genomes for paired samples was generated. Nucleotide similarity was visualized using a heatmap and dendrogram produced with the heatmap.2 function in gplots [42]. To determine breadth of genome coverage associated with sequencing effort, total reads for paired samples were downsampled to 100,000 and 10,000, using seqkit [36] prior to consensus sequence generation. Genome completeness was then subsequently compared across paired samples using the full dataset and downsampled datasets.

## Entomological and human WNV surveillance

To determine if WNV transmission dynamics varied across Nebraska during the 2023 transmission season, the state was partitioned into three regions, East, Central, and West, as described in a previous study assessing WNV seroprevalence in the state following the initial WNV epidemic in the state in 2003 [43]. These regions generally correlate with population size and average annual rainfall in Nebraska [44]. Weekly entomological surveillance data from the 2023

WNV transmission season in Nebraska was provided by the NeDHHS. WNV positivity of mosquito pools was determined by RT-qPCR [29] conducted by the NPHL. The vector index (VI), an estimate of the number of infected mosquitoes in a population [9], was calculated using the CDC PooledInfRate package available in R [45]. Total human cases of WNV per county in Nebraska were provided by the CDC National Arbovirus Surveillance System (ArboNET) [46]. The cumulative incidence statewide and for each of the three regions in 2023 was calculated by dividing the number of clinical cases of WNV in 2023 by the total population at each scale. Total WNV cases included both neuroinvasive and non-neuroinvasive cases. Population data was collected from the United States Census Bureau.

### Phylogenetic analysis of 2023 WNV genomes from Nebraska

A total of 100 coding-complete genomes were generated from Nebraska in 2023 from the West (N = 48), Central (N = 34), and East (N = 18) regions. A custom Nextstrain build [47] was generated to characterize the WNV genomes collected during the 2023 WNV transmission season in Nebraska. Briefly, complete and publicly available WNV Lineage 1A genomes and corresponding metadata were downloaded from Pathoplexus [48]. Using the augur [49] pipeline, a multisequence alignment of 3,952 WNV genomes was created using MAFFT [39]. The alignment was used as input into IQ-TREE with 1,000 ultrafast bootstraps to generate a maximum-likelihood phylogenetic tree with nodal support values [50]. A time-calibrated tree was generated and ancestral state reconstruction of clades was conducted with TreeTime [51]. A final annotated tree was visualized in Auspice. Due to the variation in entomological and human risk observed in different regions of the state, genomes from Nebraska were annotated as either East, Central, or West depending on location of collection. To determine if genotypes were shared across the state, we identified clades containing three or more Nebraska 2023 genomes, UFBoot > 70, and with a time to most recent common ancestor (tMRCA) estimated to occur in 2019 or later using the "explode tree by" function in Nextstrain. The cutoff of 2019 was selected to identify clusters made up of WNV genomes from Nebraska that could have spread between the regions during our study time period. Additionally, the majority of publicly available WNV genomes from outside of Nebraska were generated prior to 2019 (**S2 Fig**). This approach would identify WNV genotypes that may have emerged in one region and were subsequently transmitted to other regions in the state during the 2023 transmission season. The discrete ancestral state reconstructions of these clades were inferred using TreeTime [51]. Phylogenetic trees presented in this study were visualized using baltic [52].

Primer sets for the IDT xGen WNV panel are available via IDT under CS#870 for the Illumina panel and CP#1082 for the ONT panel. Note: the IDT xGen WNV panel is for research use only. Unless otherwise agreed to in writing, IDT does not intend these products to be used in clinical applications and does not warrant their fitness or suitability for any clinical diagnostics use. The purchaser is solely responsible for all decisions regarding the use of these products and any associated regulatory or legal obligations. RUO25–3814_001.

## Results

### Characterization of genomes produced by the IDT xGen WNV panel

To validate the sequencing approach, we generated libraries from 287 WNV positive mosquito pools collected during 2012–2024 in Nebraska (**Fig 1 and S3 Fig**). From these libraries, we were able to generate 274 WNV genomes with >80% genome completeness (**Fig 1A**). Of these 274 genomes, 263 had > 95% genome completeness. Mosquito pools that produced lower CT values (i.e., higher WNV RNA concentration) produced more complete WNV genomes (**Fig 1A**). Samples that produced incomplete WNV genomes (N = 10) had high rates of off-target amplification as determined by the percent of reads in the library that aligned to the WNV genome (**Fig 1A**). We observed this phenomenon in high CT value samples with low genome completeness. These samples produced quantifiable libraries and total read counts comparable to low CT values samples with high genome completeness, however the majority of reads did not align to the WNV genome. We then assessed the temporal signal of WNV genomes produced using the IDT xGen WNV panel. We observed an expected "clock-like" mutation-rate from the 274 genomes where the genetic distance was strongly

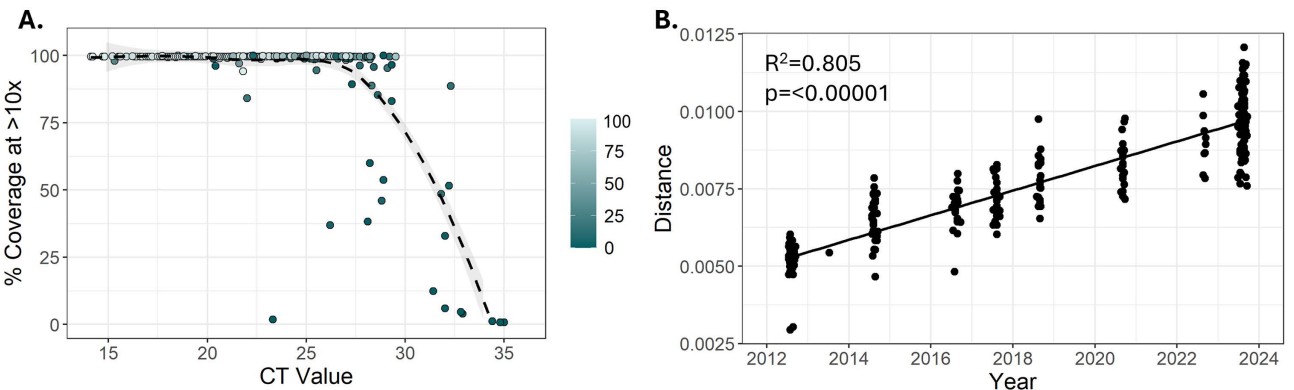

**Fig 1. Validation of WNV genomes generated using the IDT xGen WNV panel. (A)** Genome completeness for 287 genomes generated from Nebraska as measured by the proportion of positions in the WNV coding sequence with ≥ 10 reads. The color gradient for the circles depicts the percent of total reads aligning to the WNV genome. Lighter circles have a higher overall alignment rate. **(B)** Root-to-tip plot of 274 WNV genomes generated from Nebraska with >80% completeness plotting the time the sample was collected compared to the genetic distance of the sample.

correlated with sampling time (r = 0.805, p=<00001) (**Fig 1B**). From these data, we estimated the nucleotide substitution rate to be 3.97 x 10$^{-4}$ substitutions per site per year.

## Comparison of the IDT xGen WNV panel to the WNV primal scheme assay

To determine the accuracy of genomes produced with the IDT xGen WNV panel, we resequenced 30 samples using the WNV Primal Scheme assay [24] to determine if both approaches produced identical consensus sequence genomes. We observed a high level of consensus sequence genome identity between approaches across a broad range of contemporary WNV genetic diversity (**Fig 2A**). Assessment of pairwise nucleotide identity revealed that 28 of the 30 genomes sequenced using both approaches produced from the same sample were 100% identical across the complete coding sequence (**Fig 2B**). The two samples that were not identical across the genome, UNMC0019 and UNMC0567, had a total of 2 and 6 nucleotide mismatches, respectively.

We then compared genome completeness of paired samples with varying levels of sequencing effort by systematically downsampling the number of reads in each library (**Fig 2C**). At the maximum sequencing depth (i.e., the number of reads produced by a given library), both sequencing approaches produced coding-complete or nearly coding-complete genomes, with the exception UNMC0019 sequenced with Primal Scheme, which was ~75% complete. Notably, UNMC0261 showed amplicon dropout in the 10,000 read count datasets for both approaches, however the dropout was sharper for the genome generated using the IDT xGen WNV panel. The variation in genome completeness when comparing the maximum sequencing effort datasets to the most reduced datasets of 10,000 was wider for samples produced with the Primal Scheme assay compared to those produced with the IDT xGen WNV panel (**Fig 2D**).

## Comparison of the IDT xGen WNV panel for ONT MinION sequencing

A subset of these samples were resequenced using the IDT xGen WNV kit for ONT MinION sequencing. Of the 10 samples sequenced with all 3 approaches/strategies (IDT xGen WNV on Illumina, IDT xGen WNV on the ONT MinION, and WNV Primal Scheme) all produced complete and identical consensus sequence genomes (**S1 Table**).

## Entomological and epidemiological surveillance of the 2023 WNV transmission season

Entomological surveillance for WNV occurred in 21 counties across Nebraska during epidemiological weeks 22 (first week in June) to 39 (last week in September) (**Fig 3A**). Over the course of the 2023 transmission season, a total of 38,051

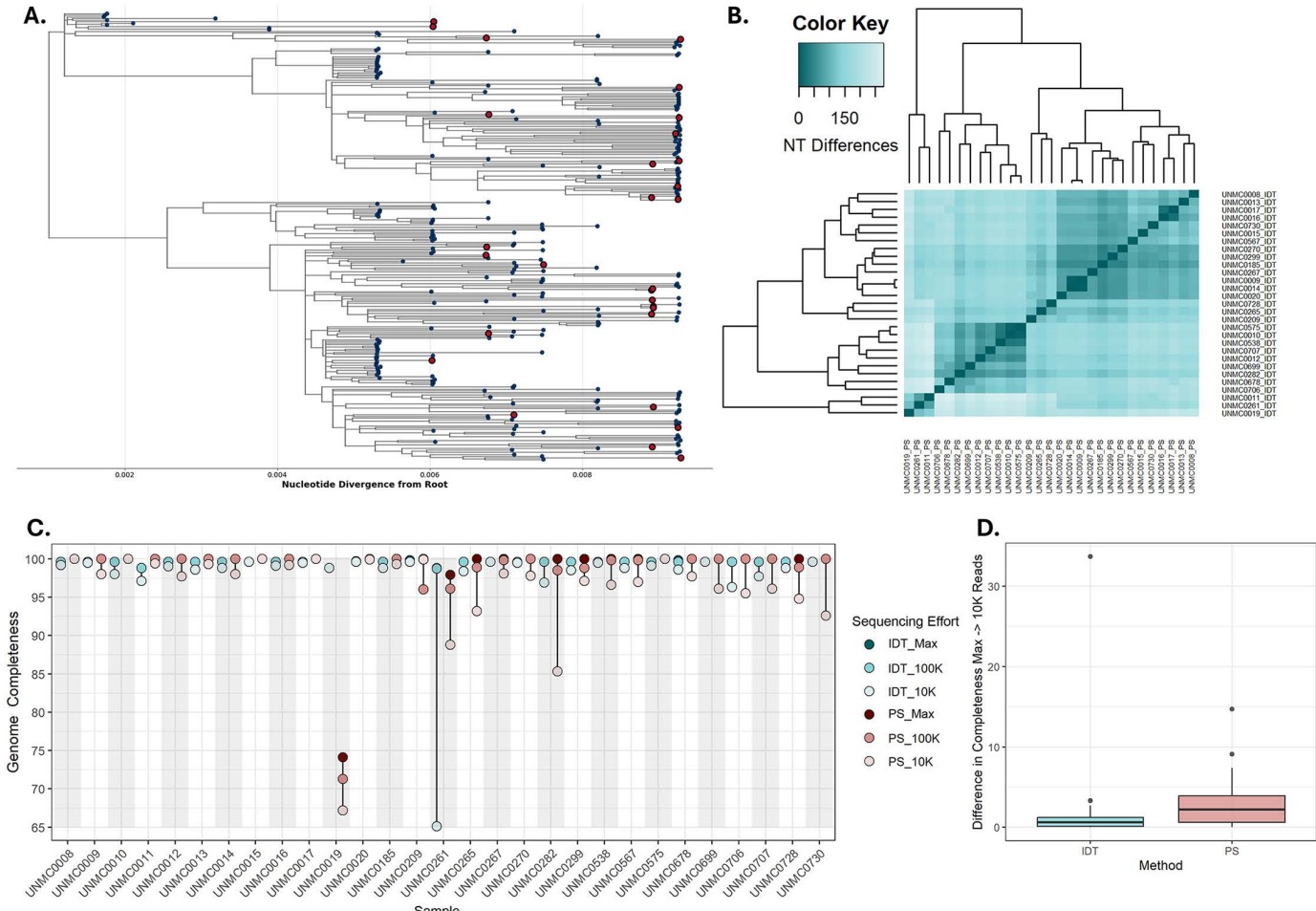

**Fig 2. Comparison of WNV genomes by sequencing approach. (A)** A maximum likelihood phylogenetic tree depicting the WNV genomes from Nebraska generated as a part of this study. Tips shown in red were sequenced using both the IDT xGen WNV panel and the Primal Scheme for WNV assay. **(B)** A dendrogram and heatmap showing the pairwise nucleotide identity of WNV genomes from the same sample generated with both approaches. **(C)** Genome completeness for each paired sample sequenced using both approaches. Red dots represent genomes sequenced with Primal Scheme and blue dots represent genomes sequenced with IDT. Sequencing effort refers to how read datasets were down sampled, fewer reads are depicted with lighter colors. Max refers to the total amount of read data for that given sample, 100K refers to 100,000 reads, and 10K refers to 10,000 reads. The lowest read count is displayed most forward. **(D)** Box-and-whisker plot showing the difference in genome completeness by sequencing approach from the max sequencing effort compared to the most reduced read dataset of 10,000 reads. Horizontal black lines represent the median difference. For B,C, and D, "PS" refers to "Primal Scheme" and "IDT" refers to "IDT xGen WNV".

*Culex* spp. mosquitoes were collected and tested for the presence of WNV RNA which resulted in 224 positive mosquito pools. The West region of the state produced the largest number of *Culex* spp. mosquitoes and mosquito pools, followed by the Central and then East regions (**Fig 3B**). The West region also had an earlier peak in mosquito abundance in epidemiological week 30 compared to epidemiological 34 in the Central region and 36 in the East region. This time difference was reflected in the vector index (VI) across each region, which was highest and rises earlier in the west compared to the rest of the state (**Fig 3C**). In total, 150 clinical cases of WNV were diagnosed in Nebraska in 2023, resulting in an overall cumulative incidence of 7.54 cases per 100,000 individuals (**Table 1**). The geographic distribution of human surveillance data reflected entomological surveillance data across the state, where the West region had a cumulative incidence of 43.72 per 100,000, which far exceeded that of the East and Central regions.

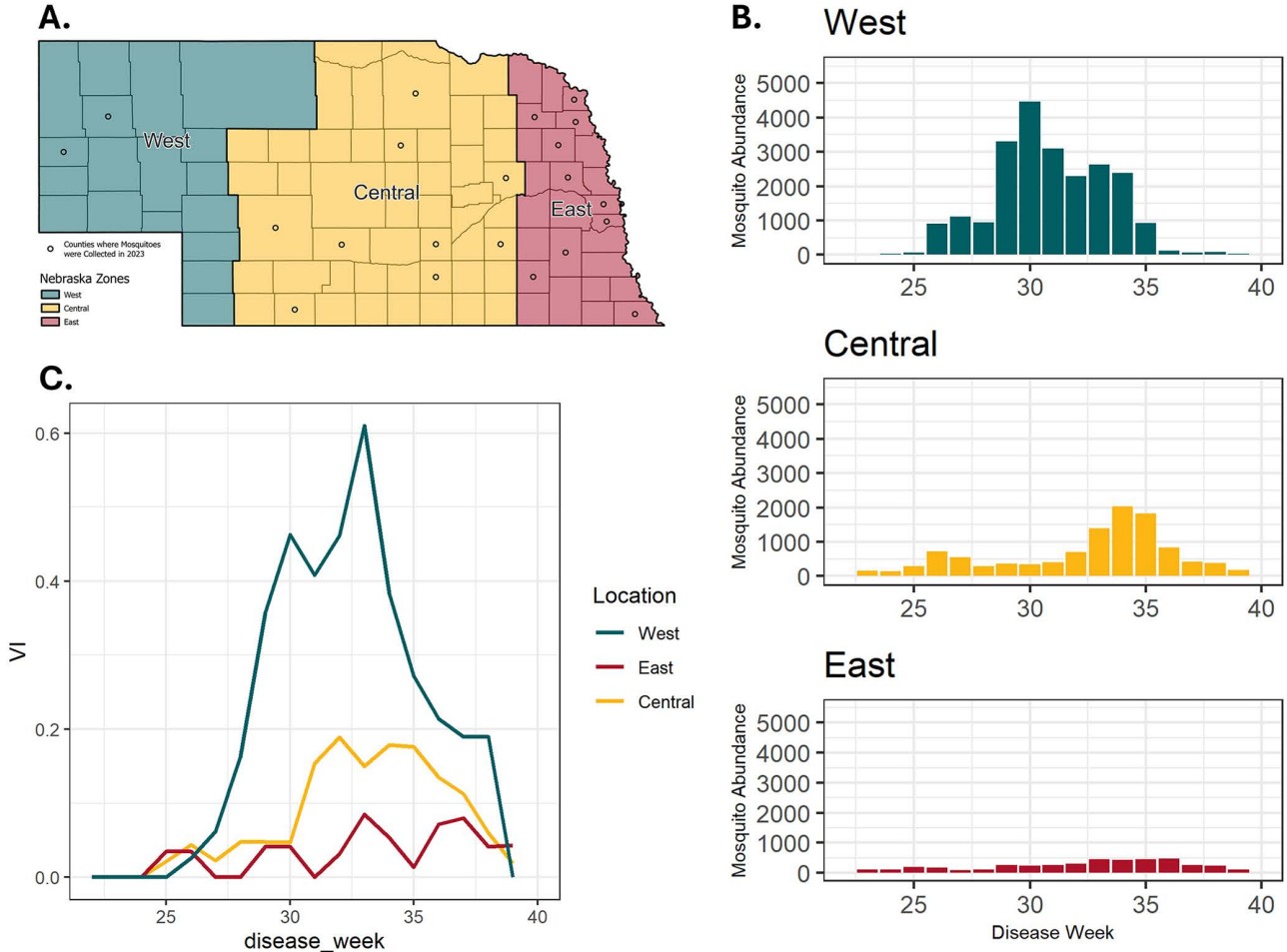

**Fig 3. Entomological description of the WNV outbreak in Nebraska in 2023. (A)** Map of Nebraska split into three zones, East, Central, and West. Counties where mosquitoes were collected and tested for WNV contain dots. Fig 3a was made in ArcGIS and the base layer was obtained from the Geographic Information Office for the State of Nebraska. Link here: https://www.arcgis.com/home/item.html?id=f8f32ae0dd254524a81477c656e3e469 **(B)** Mosquito abundance over the course of the 2023 WNV transmission season by zone in Nebraska presented by epidemiological week. **(C)** VI calculations for each zone in Nebraska over the course of the 2023 WNV transmission season presented by epidemiological week.

## Phylodynamics of the 2023 WNV transmission season

The phylogenetic analysis was focused on genomes generated from the 2023 WNV transmission season in Nebraska. Given the clear differences in entomological and human risk identified across the state (**Fig 3 and Table 1**), we aimed to determine if WNV genotypes identified during the initiation of transmission in the West region, which had the highest VI and the earliest peak in transmission, were identified in the East or Central region. A total of 100 coding-complete or nearly coding-complete WNV genomes were generated from Nebraska with sequencing effort per region proportional to the number of WNV positive mosquito pools (**Fig 4A**). All positive mosquito pools were classified as WNV Lineage 1A. The majority of genomes sequenced were singletons on long branches that extended from clades with a MRCA that did not originate in Nebraska according to our ancestral state reconstruction. To determine whether WNV genotypes were shared across the state, we identified and annotated clades that contained **1)** >3 taxa, **2)** had UFBoot values >0.7, and **3)** had a tMRCA post 2018. A total of six clades met these criteria (**Fig 4B**). The majority of samples contained within these

**Table 1. Number of cases and cumulative incidence of WNV across Nebraska.**

| Locality | Total Cases | Cumulative Incidence per 100,000 |
|---|---|---|
| **Nebraska** | **150** | **7.54** |
| East | 63 | 4.28 |
| Central | 41 | 9.95 |
| West | 46 | 43.72 |

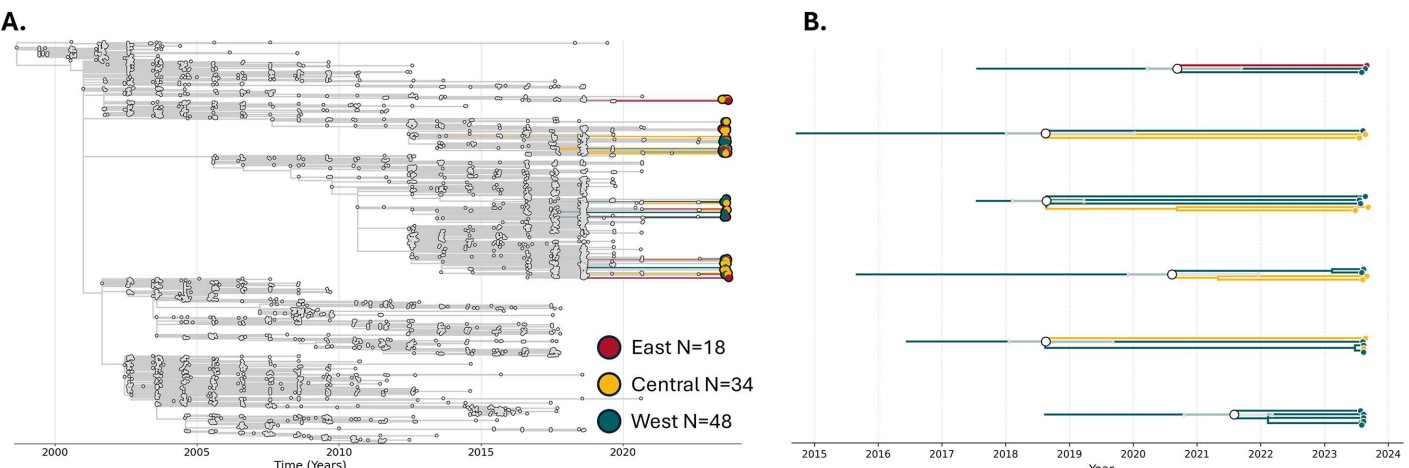

**Fig 4. Phylogenetic analysis of the 2023 WNV outbreak in Nebraska. (A)** A time-calibrated maximum likelihood phylogenetic tree highlighting 100 WNV genomes generated from Nebraska in 2023. **(B)** Exploded tree view of Nebraska WNV clades identified in 2023. Confidence intervals for the tMRCA values are represented in grey.

clades were collected in either the West or Central regions, indicating that genotypes from the eastern side of the state were unlikely to be introduced from the West region. Additionally, all clades had tMRCA values prior to 2023, indicating these genotypes had been circulating in the state prior to the 2023 transmission season. The ancestral state reconstruction indicated that all six clades had originated in the West region in Nebraska, however this could be due to sparse sampling of WNV genomes from elsewhere in the United States between 2019 and 2023 (**S2 Fig**).

## Discussion

Incorporating genomics into routine entomological surveillance will improve the understanding of WNV transmission dynamics at local and regional scales. To this end, we sought to develop and validate a single reaction amplicon-based NGS approach to generate coding-complete and accurate WNV genomes. We generated 274 coding-complete WNV genomes from mosquito pools collected in Nebraska during 2012–2024 (**Fig 1**). The IDT xGen WNV panel generated coding-complete genomes from the majority of samples assessed. Genome completeness was reduced in samples with higher WNV CT values (lower viral input quantities) as seen in other amplicon-based sequencing assays [53,54]. Interestingly, samples with high CT values that failed to produce coding-complete genomes had relatively equal read distributions across regions of the WNV genome but had high rates of off-target read alignment, indicating that increasing the amount of sequence data from such samples would result in more complete genomes. WNV genomes from Nebraska demonstrated an expected clock-like mutation-rate and an inferred nucleotide substitution rate of $3.97 \times 10^{-4}$ substitutions per site per year is on par with estimates from other WNV studies [11,19,20,22,55].

In addition to genome completeness, accuracy of genomic content is crucial for correct phylogenetic inference.

For 30 samples representing a diversity of WNV genotypes across Nebraska, we compared consensus sequence genomes generated using the IDT xGen WNV panel and the WNV Primal Scheme assay (**Fig 2**). A pairwise comparison demonstrated 100% nucleotide identity across the WNV genome in 28 of the 30 paired consensus sequence genomes. The two samples that did not produce 100% identical genomes, UNMC0019 and UNMC0567, had 2 and 6 nucleotide mismatches, respectively. Upon further inspection of these alignments, nucleotide mismatches were the result of an ambiguous nucleotide (e.g., "Y" to represent either a "C" or "T") called in the genomes generated with the WNV Primal Scheme assay. However, as the Primal Scheme bioinformatic pipeline was originally set to call a base with a simple majority frequency of >50%, modifying the consensus sequence generation step to allow for ambiguous nucleotides resolved this discrepancy. Additionally, amplicon dropout that resulted in reduced genome completeness occurred sparingly in both approaches, and downsampled datasets that contained only 10,000 reads produced coding-complete or nearly coding-complete genomes in both sequencing approaches.

While both approaches produced high-quality genomes, there are important distinctions between the two. For one, the IDT xGen WNV panel has a streamlined laboratory protocol where amplicon generation occurs in a single tube per sample, whereas two separate PCRs are required for Primal Scheme assays [24,25]. Additionally, the IDT xGen WNV panel allows for amplicon generation and library preparation for both Illumina and Oxford Nanopore MinION sequencing platforms from a single kit. The Primal Scheme approach, however, offers more flexibility in reagents for amplicon generation and library preparation. The IDT xGen WNV panel encompasses the full WNV CDS, where the Primal Scheme assay includes targets in both the 5' and 3' UTR, resulting in greater proportion of the total genome represented in final consensus sequences. Incorporating primer sequences to target the UTRs is an important future direction for the IDT xGen WNV panel.

The 2023 WNV transmission season in Nebraska was particularly severe with 150 cases of WNV disease diagnosed across the state, the most on record since 2018 when 251 cases were diagnosed (**Table 1**) [2]. Nebraska had the fourth most reported WNV cases in 2023 behind Colorado, California, and Texas, respectively [56]. Analysis of entomological data, specifically the abundance of mosquitoes and the VI, indicated that enzootic transmission of WNV was more severe in the West region of the state compared to the Central and East regions (**Fig 3**). This pattern was reflected in human case data in different regions across the state, where the cumulative incidence is substantially higher in the West region, 43.72 per 100,000, compared to the Central and Eastern regions, 9.95 and 4.28 per 100,000, respectively. These results demonstrate that entomological risk metrics, such as the VI, were useful to predict human cases at large geographical scales, as observed elsewhere [8].

In addition to a higher abundance of *Culex* mosquitoes and a higher VI, enzootic transmission peaked earlier in the West region. This observation led us to explore if earlier and more intense transmission in the West region resulted in dispersal of WNV to the rest of the state. To answer this question, we generated 100 coding-complete WNV genomes from WNV positive mosquito pools collected across the state in 2023 using the IDT xGen WNV panel (**Fig 4**). We demonstrated that WNV genomes from Nebraska sequenced in 2023 were broadly distributed across the WNV Lineage 1A phylogeny, suggesting little association between genotype and region of collection. A similar study assessing WNV diversity from Suffolk County, NY in 2012 also found high levels of WNV sequence diversity from samples collected in the same location [57]. Additional studies have also found substantial genetic diversity in sequenced WNV genomes from the same locations in the same collection years [58–60].

To explore this question further, we conducted a discrete phylogeographic analysis using a time-calibrated maximum-likelihood phylogenetic tree to identify clades that contained genomes from across the state that would indicate shared transmission networks. The results of our phylogenetic assessment indicated that of the 100 WNV genomes generated from Nebraska in 2023, six clades containing 26 genomes met our criteria for nodal support and tMRCA. Each of these clades had a > 95% discrete state probability of arising from Nebraska and specifically in the West region. A single genome derived from samples collected in the East region was placed in a clade with genomes from the West region, where every

other clade contained genomes from either the West and Central regions, or the West region alone. No clades had an estimated tMRCA in 2023 and most had tMRCA estimates prior to 2021, suggesting these genomes have been present in the state over multiple transmission seasons. In general, these results suggest an absence of geographic structure of WNV genomes by broadly defined regions in Nebraska during the 2023 outbreak. A study of WNV dynamics in New York state by Bialosuknia et al. found a similar absence of geographic structure in their phylogenetic analysis with the exception of two well-supported clades [22], one of them being the NY10 genotype which was associated with an increase in cases in 2012 [18,21]. The persistence of specific WNV genotypes in high-incidence areas in Nebraska remains to be determined. Future work will focus on longitudinal genomic analyses to evaluate how and why genotypes persist in the state once more complete genomic datasets are generated.

Multiple limitations of this study need to be considered. First, our comparative analysis of the IDT xGen WNV panel to the Primal Scheme assay for WNV was limited to samples collected from Nebraska. Although these genomes do represent substantial contemporary WNV genetic diversity (Fig 2A), circulating genotypes such as SW03 were not compared. Additionally, our phylogeographic analysis was limited to a snapshot of the 2023 WNV transmission season in Nebraska. While we were able to generate numerous genomes from 2023, the lack of clustering of Nebraska samples from this year may be due to insufficient sampling in our sequencing data. Thus, we would likely need to sequence more genomes from 2023 to observe clustering patterns. In addition, the genomes from the 2023 outbreak in Nebraska were on long branches extending out from well supported clades, likely because there was a lack of contextual WNV sequence data from elsewhere in the United States for the same time period (S2 Fig). A temporal analysis from Nebraska as more data becomes available from around the United States will provide better estimates of clustering patterns of WNV genomes in high-incidence settings.

In conclusion, our results indicated that the IDT xGen WNV panel produces high-quality WNV genomes that can be used to characterize WNV transmission dynamics.

## Supporting information

**S1 Fig. Cartoon schematic highlighting the differences in amplification approaches between Primal Scheme (A) and the WNV IDT xGen approach (B).**
(TIF)

**S2 Fig. Publicly available WNV genomes used in this study by year or collection.**
(TIF)

**S3 Fig. WNV Genomes generated as a part of this study by year of collection.**
(TIF)

**S1 Table. Comparison of sequencing data and consensus sequence genomes from 10 matched samples prepared and sequenced using different approaches.**
(XLSX)

## Acknowledgments

We would like to acknowledge Nathan Grubaugh from the Yale School of Public Health for helpful comments on analysis and providing the Primal Scheme primer pools. We would also like to acknowledge the national arbovirus surveillance system ArboNET for providing human WNV case data for the state of Nebraska for 2023.

## Author contributions

**Conceptualization:** Shawn Freed, Nick Downey, Kaylee S. Herzog, Joseph R Fauver.

**Data curation:** Joseph R Fauver.

**Formal analysis:** Shawn Freed, Sarah Chandler, Sarah Uhm, Zach Pella, Kaylee S. Herzog, Joseph R Fauver.

**Funding acquisition:** Joseph R Fauver.

**Investigation:** Shawn Freed, Sarah Chandler, Sarah Uhm, Zach Pella, Dikchha Gurung, Halie Smith, Tammy Dowdy, Amanda M. Bartling, Ava Butz, M. Jana Broadhurst, Sydney Stein, Emily L. McCutchen, Jeff Hamik, Peter C. Iwen, Kaylee S. Herzog, Joseph R Fauver.

**Methodology:** Michael R. Wiley.

**Supervision:** Joseph R Fauver.

**Visualization:** Joseph R Fauver.

**Writing – original draft:** Shawn Freed, Joseph R Fauver.

**Writing – review & editing:** Sydney Stein, Emily L. McCutchen, Jeff Hamik, Peter C. Iwen, Nick Downey, Kaylee S. Herzog, Joseph R Fauver.

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
