## [Decision Letter · Decision Letter 0]

23 Dec 2025

Response to Reviewers
Revised Manuscript with Track Changes
Manuscript

Shaden Kamhawi

co-Editor-in-Chief

Paul Brindley

co-Editor-in-Chief

**Journal Requirements:**

At this stage, the following Authors/Authors require contributions: Shawn Freed, Sarah Chandler, Sarah Uhm, Zach Pella, Dikchha Gurung, Halie Smith, Tammy Dowdy, Amanda M. Bartling, Ava Butz, Michael R. Wiley, M. Jana Broadhurst, Sydney Stein, Emily L. McCutchen, Jeff Hamik, Peter C. Iwen, Nick Downey, Kaylee S. Herzog, and Joseph R Fauver. Please ensure that the full contributions of each author are acknowledged in the "Add/Edit/Remove Authors" section of our submission form.

3) We notice that your supplementary Figures, and Tables are included in the manuscript file. Please remove them and upload them with the file type 'Supporting Information'. Please ensure that each Supporting Information file has a legend listed in the manuscript after the references list.

**Reviewers' comments:**

**Key Review Criteria Required for Acceptance?**

**Methods:**

-Are the objectives of the study clearly articulated with a clear testable hypothesis stated?

-Is the study design appropriate to address the stated objectives?

-Is the population clearly described and appropriate for the hypothesis being tested?

-Is the sample size sufficient to ensure adequate power to address the hypothesis being tested?

-Were correct statistical analysis used to support conclusions?

-Are there concerns about ethical or regulatory requirements being met?

Reviewer #1: (No Response)

Reviewer #2: -Are the objectives of the study clearly articulated with a clear testable hypothesis stated? Yes

-Is the study design appropriate to address the stated objectives? Yes

-Is the population clearly described and appropriate for the hypothesis being tested? Yes

-Is the sample size sufficient to ensure adequate power to address the hypothesis being tested? Yes

-Were correct statistical analysis used to support conclusions? N/A

-Are there concerns about ethical or regulatory requirements being met? No

**Results**

-Does the analysis presented match the analysis plan?

-Are the results clearly and completely presented?

-Are the figures (Tables, Images) of sufficient quality for clarity?

Reviewer #1: (No Response)

Reviewer #2: -Does the analysis presented match the analysis plan? Yes

-Are the results clearly and completely presented? Needs minor revisions (see below)

-Are the figures (Tables, Images) of sufficient quality for clarity? Needs minor revisions (see below)

**Conclusions**

-Are the conclusions supported by the data presented?

-Are the limitations of analysis clearly described?

-Do the authors discuss how these data can be helpful to advance our understanding of the topic under study?

-Is public health relevance addressed?

Reviewer #1: (No Response)

Reviewer #2: -Are the conclusions supported by the data presented? Yes

-Are the limitations of analysis clearly described? Yes

-Do the authors discuss how these data can be helpful to advance our understanding of the topic under study? Yes

-Is public health relevance addressed? Yes

**Editorial and Data Presentation Modifications?**

Reviewer #1: (No Response)

Reviewer #2: MAJOR COMMENTS:

1. Line 29: Authors must be cautious when using the term “complete genomes” as the IDT panel does not cover the 5’ and 3’ UTRs (as stated in lines 119-120). Authors must rewrite every instance of “complete genome” in the manuscript to “coding-complete”, which is the standard language used when genome termini are not fully determined.

2. Line 30: While the IDT xGen WNV panel undoubtedly produces coding-complete genomes, I am not convinced that the authors provide significant evidence to say that the panel is “more robust to amplicon dropout”. Although theoretically, super amplicons generated with the IDT xGen panels are expected to provide coverage when nucleotide variants occur in a primer binding site, the data presented in this manuscript do not demonstrate that this happens in any meaningful way using either sequencing approach. In fact, the only example given, that of sample UNMC0261, shows “amplicon dropout” with both methods, with the most significant dropout present in the genome generated with the IDT xGen panel. The authors themselves highlight this (lines 277-278 and line 375); therefore, I am unsure why this claim was left in line 30 of the abstract, but the authors should consider revising this sentence.

3. Line 122: “Sample selection” section needs significant clarification on the methodology. Do all 287 WNV-positive pools have the same number of individuals per pool? How were insect pools generated? Provide details on logistical workflow from field to lab (i.e., cold chain maintained? Were they stored in stabilization buffer such as RNAlater or similar?). How was species identification performed? (morphological identification, DNA barcoding?). Please provide as much detail as possible.

4. Line 131: Were pools mechanically disrupted prior to extraction? (i.e., using a tissue homogenizer, mortar and pestle, etc.?) Please be as detailed as possible.

5. Line 137: How much cDNA was used as input for the IDT Amplicon Core kit?

6. Line 165: Reference 25 is only for the PrimalScheme open-source preprint, not a specific protocol designed for WNV sequencing. Please correct, as this is misleading.

7. Line 318, Figure 3: Distribution of the 224 WNV-positive mosquito pools across the three regions surveilled should be included somewhere in figure 3 or at least mentioned in the results. Only the total is given for the whole state, followed by which region produced the largest number of Culex spp. pools.

8. Line 256, Figure 1A: The Figure is slightly confusing. Data is very informative and fine as is, but the Y-axis needs more explanation in the figure legend. I assume that the Y-axis represents the percentage of the genome sequence with a coverage depth of greater than 10x (at least 10 reads at each base), but this is not clearly explained anywhere in the manuscript.

9. Lines 148-153 and 176-180: Providing more detailed information on how the reads were quality controlled (quality score limits, removal of terminal nucleotides, minimum read length, etc.) and final genome read coverage (average depth of coverage) would significantly improve this manuscript.

10. Line 231: As currently written, I do not believe the data availability for all sequencing data generated meets the standards required for this journal. The GitHub link provided is useful, but it does not make it easy for readers to access the consensus sequences and corresponding sequencing metrics. Please edit the sample metadata table in GitHub to include all relevant metadata associated with each generated genome, including columns for GenBank accession numbers (hyperlinked to publicly available data), total number of reads, number of reads mapped to WNV, sequence coverage %, length, GC content, etc.

11. Line 231: Additionally, the data availability section makes no mention of whether the raw sequence data will be made available through a public repository such as NCBI Sequence Read Archive (SRA), which is required (link to PLOS submission guidelines and recommended data repositories in link below, as well as two publications for reference).

https://journals.plos.org/plosntds/s/recommended-repositories

1. Palinski RM, Sangula A, Gakuya F, Bertram MR,Pauszek SJ, Hartwig EJ, Smoliga GR, Obanda V,Omondi GP,VanderWaal K, Arzt J.2022.Genome Sequences of Foot-and-Mouth Disease Virus SAT1 Strains Purified from Coinfected Cape Buffalo in Kenya. Microbiol Resour Announc11:e00584-22.https://doi.org/10.1128/mra.00584-22

2. He X, Yin Q, Zhou L, Meng L, Hu W, et al. (2021) Metagenomic sequencing reveals viral abundance and diversity in mosquitoes from the Shaanxi-Gansu-Ningxia region, China. PLOS Neglected Tropical Diseases 15(4): e0009381. https://doi.org/10.1371/journal.pntd.0009381

MINOR COMMENTS:

1. Line 86: References needed for prior studies. Examples of “important factors”.

2. Line 98: No need to overstate by saying that other approaches require “multiple” PCR reactions, while (see line 112) in fact it is only two separate PCR reactions. This does not diminish the usefulness of using a single-tube approach.

3. Line 99: Though not grammatically incorrect, authors should revise every instance of “as well,” at the beginning of a sentence and perhaps substitute it with “Moreover” or “In addition”.

4. Line 149: “adapter” is the preferred spelling when referring to Illumina adapter sequences.

5. Line 168: eluted with 25uL of what?

6. Line 176: Same suggestion as in line 149.

7. Line 286: “genomes” is misspelled.

8. Line 407: The word “sequence” is repeated.

**Summary and General Comments**

Reviewer #1: This study validates a single-reaction amplicon-based sequencing method for generating high-quality WNV genomes. The method was compared to the commonly used Primal Scheme assay and found to be robust and efficient, especially in handling amplicon dropout. Using this approach, they sequenced >250 complete WNV genomes from mosquito surveillance samples collected in Nebraska. Phylogenetic analysis revealed high genetic diversity with minimal geographic clustering of virus strains in the state during 2023. The genomic sequences and development of a new sequencing approach provides a valuable resource and methodology for the research community. Overall, the paper reads well, the data is clearly presented and is appropriately interpreted in the context of the larger literature. I have only minor comments for consideration.

Specific Comments:

1. Abstract, line 30. It states they generated 100 complete genomes in 2023 but in the manuscript >250 genomes were produced from multiple years. I think it’s worth noting the full extent of their accomplishment in the abstract.

2. Figure 4 is low resolution and could be made larger for easier assimilation.

3. Discussion, line 412 not sure what is meant by “shared transmission cycles”. WNV is maintained in a bird-Culex transmission cycle throughout its geographic range. Perhaps change to “shared transmission networks”?

Reviewer #2: In this work, Freed et al. validate a single-reaction, amplicon-based approach for sequencing coding-complete West Nile virus genomes from mosquito pools collected during routine WNV vector surveillance. The protocol is based on a custom IDT xGen WNV panel designed in collaboration with IDT. The authors use a subset of samples to compare this method to the Primal Scheme approach, a previously established, amplicon-based sequencing protocol commonly used for sequencing WNV. The authors provide sufficient evidence to demonstrate that the IDT xGen WNV panel can generate coding-complete WNV genome sequences, the majority of which exhibit 100% nucleotide identity with those generated by the Primal Scheme assay.

Finally, the authors demonstrated the utility of this validated IDT xGen WNV sequencing protocol by generating coding-complete WNV genomes from 100 mosquito pools collected during the 2023 transmission season in Nebraska. They then used this sequence data to characterize WNV transmission dynamics in this region.

Overall, this work is relevant to researchers interested in implementing a WNV genome sequencing protocol using an efficient approach that uses a single-tube method. The manuscript is well-written and organized, with figures that clearly support its major claims.

There are minor revisions that would improve the clarity of the manuscript, particularly involving the methodology and presentation of the sequencing data. All revisions are categorized into “major comments” and “minor comments” below.

I believe the manuscript is acceptable for publication, provided the authors address these revisions.

PLOS authors have the option to publish the peer review history of their article (what does this mean? ). If published, this will include your full peer review and any attached files.

**Do you want your identity to be public for this peer review?** For information about this choice, including consent withdrawal, please see our Privacy Policy .

Reviewer #1: No

Reviewer #2: No

**Figure resubmission:**

**Reproducibility:** To enhance the reproducibility of your results, we recommend that authors of applicable studies deposit laboratory protocols in protocols.io, where a protocol can be assigned its own identifier (DOI) such that it can be cited independently in the future. Additionally, PLOS ONE offers an option to publish peer-reviewed clinical study protocols. Read more information on sharing protocols at https://plos.org/protocols?utm_medium=editorial-email&utm_source=authorletters&utm_campaign=protocols

---

## [Editor Report · Decision Letter 1]

13 Jan 2026

Dear Dr. Fauver,

We are pleased to inform you that your manuscript 'Genomic Characterization of a Severe West Nile Virus Transmission Season using a Single Reaction Amplicon Sequencing Approach' has been provisionally accepted for publication in PLOS Neglected Tropical Diseases.

Best regards,

Doug E Brackney, PhD

Academic Editor

Nigel Beebe

Section Editor

Shaden Kamhawi

co-Editor-in-Chief

Paul Brindley

co-Editor-in-Chief

---

## [Editor Report · Acceptance letter]

Dear Dr. Fauver,

We are delighted to inform you that your manuscript, "Genomic Characterization of a Severe West Nile Virus Transmission Season using a Single Reaction Amplicon Sequencing Approach," has been formally accepted for publication in PLOS Neglected Tropical Diseases.

Best regards,

Shaden Kamhawi

co-Editor-in-Chief

Paul Brindley

co-Editor-in-Chief
